# Access to support services for individuals who experience intimate partner violence during stressful life events (SLEs) in high-income countries: Protocol for a scoping review

Dina Idriss-Wheeler[1]*, Ziad El-Khatib[2,3,4], Sanni Yaya[5,6]

1 Interdisciplinary School of Health Sciences, University of Ottawa, Ottawa, Canada, 2 Department of Global Public Health, Karolinska Institutet, Stockholm, Sweden, 3 University of Global Health Equity (UGHE), Kigali, Rwanda, 4 World Health Programme, Université du Québec en Abitibi-Témiscamingue (UQAT), Rouyn-Noranda, Quebec, Canada, 5 School of International Development and Global Studies, University of Ottawa, Ottawa, Canada, 6 The George Institute for Global Health, Imperial College London, London, United Kingdom

* didri040@uottawa.ca

**Data Availability Statement:** No datasets were generated or analyzed during the current study.

## Abstract

### Background

Women, gender minorities and their children are at heightened risk of intimate partner violence (IPV) following stressful life events (SLE). The increase in IPV during the global pandemic of the Novel Coronavirus (COVID-19) is recent evidence. Studies have linked IPV to poor health, resulting in lower mental, physical, sexual, and reproductive health outcomes. IPV has also been shown as a barrier to labour force participation, leading to negative socio-economic outcomes (i.e., low or no employment). Formal and informal supports help individuals who experience IPV, but it is unclear if and how these are being accessed during SLEs such as environmental disasters, pandemics, and economic recessions. Accessibility to programs is an issue in normal times because of stigma, social norms, and lack of knowledge; this has been further amplified by situations where individuals who experience violence are isolated physically and emotionally, as well as face controlling behaviours by their perpetrators of violence. This scoping review will be used to conduct a comprehensive review of literature and address the research question: *What is known in published literature about access to services by individuals who experience IPV during stressful life events in high-income countries*?

### Methods

The following electronic databases will be searched for relevant publications: MEDILINE (OVID), Embase (OVID), PsychINfo (OVID), CINAHL (EBSCO), Global Health (EBSCO), Gender Watch (ProQuest), Web of Science and Applied Social Sciences Index & Abstracts (ProQuest). Key terms and medical subject headings (MeSH) will be based on previous literature and consult with an expert librarian. The major concepts include 'stressful life events' AND intimate partner violence' AND 'access to services'. Google, Google Scholar, and the

**Funding:** The author(s) received no specific funding for this work.

**Competing interests:** The authors have declared that no competing interests exist.

WHO website will be used to search for grey literature, books/chapters, and programme reports as well as references of relevant reviews. Studies will be screened and extracted by two reviewers and conflicts resolved through discussion or a third reviewer. Both quantitative and qualitative analysis of relevant data will outline key findings.

## Discussion

The scoping review will provide synthesized and summarized findings on literature regarding access to informal and formal social supports by victims of IPV during SLEs (i.e., pandemics and natural/environmental disasters/emergencies, economic recessions) where possible, highlighting key barriers, facilitators and lessons learned. Findings have potential to inform programs, policies, and interventions on accessibility to necessary support and health services during disasters.

## Background

Intimate partner violence (IPV) is "any behaviour within an intimate relationship that causes physical, psychological or sexual harm to those in the relationship" [1]. This includes controlling behaviours that lead to isolation from family and friends, monitoring movements, as well as restricting access to financial resources, employment, education, social services, or medical care. Isolation and taking away the freedoms of the victims to maintain power and control are paramount forms of IPV during stressful life events (SLEs).

SLEs are "undesirable, unscheduled, nonnormative, and/or uncontrollable discrete, observable events with a generally clear onset and offset that usual signify major life changes" [2]. In the context of IPV, public health emergencies, disasters and economic recessions are known as key SLEs that lead to increased risk of or exacerbation of existing IPV. A recent systematic review on domestic violence during COVID-19 revealed that several countries reported increased violence post-lockdown [3], most likely due to victims isolated with their abusive partners. Historically, studies since the Great Depression have outlined the harmful effects of economic uncertainty on marital conflict and quality, particularly looking at the effects of unemployment, loss of income and economic hardship [4–6]. Schneider et al., (2016) revealed that rapid increases in unemployment rates increased male partner's controlling behaviour, even after controlling for unemployment and economic stress. Findings suggested that anxiety and uncertainty have negative effects on relationships [6]. A recent systematic review assessed the global literature on the association between disasters from natural hazards and violence against women and girls (VAWG), concluding positive associations between disaster exposure and increased VAWG [7].

It is well documented in literature that IPV is a population health problem that leads to health inequities faced by survivors globally; it is linked with poor psychological, physiological and behavioural health outcomes [1, 8]. A 2018 systematic review and meta-analysis of cohort studies investigated the association between IPV and adverse health outcomes or health risk behaviours [9]. Findings revealed a positive statistically significant relationship between depressive symptoms and subsequent IPV as well as a bidirectional relationship between recent IPV and substance use [9]. Long-term outcomes of the abuse are now emerging with a recent study that focused on the long-term effects of sexual abuse, indicating a higher risk of developing brain damage linked to stroke, cognitive decline and dementia [10]. Encouragingly,

access to formal and informal (perceived or received) social and health supports are beneficial to mental health, physical health and well-being [11, 12]. In fact, theories regarding factors implicated in the experience of violence and how social networks impact health are well established [13].

For individuals who have experienced IPV, the importance of *formal* (i.e. social service providers, violence against women (VAW) services, criminal justice, health professionals, religious organizations) and *informal* (i.e., friends, relatives, neighbours) supports for improved health outcomes is well documented in literature [13, 14]. Social supports, particularly informal ones, mean closer relationships (i.e., increased psychological and material resources) that lead to better mental health [15, 16]. For victims of IPV, better social supports, and even perception of supports, are a protective factor because the emotional and tangible support reduces susceptibility to negative psychological impacts of partner abuse [14]. Similarly, both formal and informal supports have been shown to protect individuals who experience physical violence [14]. Strong informal support networks have been shown to reduce the experience of less severe physical violence over the course of a year [17]. Furthermore, those who used formal services (i.e., shelters, civil protection, legal advocacy services) were less likely to experience IPV [14]. A recent systematic review revealed that advocacy and case-management interventions with strong linkages to the community improved the access to social supports (i.e., resources, coping strategies) and mental health outcomes of IPV survivors [13]. So, what happens to survivors and how do they access services when a stressful life event such as a pandemic, environmental disaster or economic recession takes place?

Previous work in the disaster arena demonstrated challenges in access to formal and informal support services to inform disaster and emergency preparedness [18]. Enarson (1999) documented the organizational readiness, impact and response by domestic violence programs in the US and Canada in communities hit by environmental disasters [19]. The study outlined a gendered lens to disaster social science. It related the issues of increased violence and vulnerability to violence during and in the aftermath of a disaster and women's diminished access to resources and supports [19]. A small study post-2008 economic downturn in Ireland suggested that the recession had a direct impact on VAW services and their funding [20]. Given that the general source of funding came from grants or fundraising, the economic downturn meant that less disposable income for donations by the community and government negatively affected the number of services available, reducing access to formal services [20]. Unfortunately, for victims of violence, the recent COVID-19 lockdown has only increased isolation from informal and formal social and health networks.

Some scoping reviews are currently investigating the impact of COVID-19, with specific focus on access to and utilization of services for sexual and reproductive health. This includes services for gender-based violence and IPV [21], while others are looking at virtual supports and IPV services [22]. There are also reviews underway on women's mental health and experience of IPV following natural disasters [23] and the impact of IPV on work and employment [24]. Additionally, research studies are looking at service provider perspectives on how the pandemic restrictions affected survivors [25] as well as their engagement with advocacy services during COVID-19 [26].

The recent COVID-19 pandemic has introduced several rapid reviews, commentaries and editorials on the experience of IPV during the pandemic with some anecdotal evidence regarding formal and informal support utilization, with preliminary recommendations to service providers working with IPV victims [27–29]. However, no review to the authors' knowledge has identified key characteristics/factors related to accessibility to services by individuals who experience IPV during SLEs. More specifically, the review will focus on

SLEs known to exacerbate or increase risk of IPV because of increased isolation, financial dependence and controlling behaviours—these include pandemics, natural disasters, and economic recessions.

## Rationale for a scoping review

A scoping review is appropriate in this case because (i) it involves synthesis and analysis of heterogeneous material on the topic, and (ii) it is looking at identifying and reporting on certain concepts and factors associated with access to services by IPV survivors during SLEs [30, 31].

It is apparent that several factors are implicated in access to services during SLEs by individuals who experience IPV, and a scoping review of the literature will clarify key concepts and contribute to existing knowledge on the accessibility of services during disasters and emergencies. This is a relevant topic considering the recent pandemic as well as the increase in climate-related natural disasters [32] and economic recessions in high-income countries (refer to classification by the World Bank Group [33] over the last two decades [34]. Furthermore, the rapidly emerging work from the current COVID-19 pandemic provides an excellent opportunity to update and supplement existing literature.

Two conceptual frameworks guide this work–Berkman and Krishna's (2014) conceptual model of how social networks impact health and Heise's (2012) heuristic-ecological model of factors associated with the experience of IPV. The former provides a framework to examine how macro social structural conditions (i.e., stressful life events) can affect the structure of social networks (i.e., formal VAW social services, informal community support services) at the mezzo level that lead to micro psychosocial mechanisms (disrupted access to services; job loss; school; experience of IPV) to take place and impact health through behavioural, psychological and physiological pathways [35]. The later uses an ecological framework to conceptualize factors at different system levels that play into the experience of violence [16]. The conceptual models are complementary and frame how socio-ecological factors at different levels (macro political/cultural/social, mezzo community, micro individual) enhance or suppress the ability to access services and impact health of IPV survivors through behavioural, psychological, and physiological pathways.

## Stage 1: Identification of the research question and objectives

The objective of this scoping review is to examine, summarize and characterize existing literature on access to informal and formal social supports by survivors of IPV during SLEs, including pandemics, natural disasters, and economic recessions in high-income countries. The aim is to understand key factors and issues implicated in access to formal and informal service provision during SLEs by survivors of IPV that can inform work or research on providing better access during these circumstances. The research question is:

*What is known in published literature about access to services by individuals who experience intimate partner violence during SLEs in high income countries*? More specifically,

- What are the barriers and facilitators to accessing informal and formal supports by victims of IPV during SLEs?

- What are the barriers and facilitators to providing informal and formal supports to victims of IPV during SLEs?

- What approaches are used for sub-groups at higher risk of isolation (i.e., gender minorities, women, girls, minority/ethnic racialized populations, and individuals living in rural remote regions)? How do these sub-groups access support during SLEs?

- What are key lessons learned to date from attempting to access or provide formal/informal supports to victims of IPV during SLEs?

## Methods

The methodological frameworks for a scoping review as described by Arksey and O'Malley (2005) and the Joanna Briggs Institute (2020) will be used [31, 36]. The Preferred Reporting Items for Systematic Reviews and Meta-Analysis extension for Scoping Reviews (PRISMA-ScR) will guide the scoping review [37]. The Preferred Reporting Items for Systematic Reviews and Meta-Analysis for Protocols (PRISMA-P) will be used to guide the development of this scoping review protocol (S1 Appendix) [38]. This review has been submitted for registration in Open Science Framework.

### Stage 2: Identifying relevant studies

In terms of information sources and search strategy, the following electronic databases will be used to search for peer-reviewed studies: MEDLINE (OVID), Embase (OVID), PsycInfo (OVID), CINAHL (EBSCO), Global Health (EBSCO), Gender Watch (ProQuest), Web of Science, and Applied Social Sciences Index & Abstracts (ProQuest). Key terms and medical subject headings (MeSH) will be based on previous literature and consultation with an expert librarian. The major concepts include 'stressful life events' AND 'intimate partner violence' AND 'access to services'. Google, Google Scholar, and the WHO website will be used to search for grey literature, books/chapters, and programme reports. Reference review of selected articles will also take place. Table 1 is the preliminary search strategy in Medline created in consultation with the University of Ottawa Population Health librarian and modified from previously completed search strategies using similar key terms [39–41]:

**Eligibility criteria.** Key inclusion and exclusion criteria for this scoping review are described in detail below and summarized in Table 2.

*Types of participants*. The types of participants in this review include individuals who experienced IPV during pandemics, natural disasters, and economic recessions in high-income countries. In many regions, and according to the WHO, the experience of IPV occurs as young as 13 [1], therefore the population will include individuals (regardless of gender), 13 or older who experienced IPV. In times of disasters, women, children and gender minorities are at heightened risk of violence, as seen recently with increased IPV reports during COVID-19 pandemic lockdowns [42, 43]. A commentary by Moffitt et al. (2020) identified the unique impact of COVID-19 on experience of abuse for both victims and the service providers living in rural, remote and northern communities where IPV and femicide pre-pandemic are higher than in larger cities [44]. Where relevant, a sub-group analysis of these populations will be considered.

*Concepts*. The concept or phenomenon of interest being studied in this scoping review is access to informal or formal supports by individuals who experience IPV during stressful life events such as public health emergencies, disasters, and economic recessions. Supports are defined as <u>formal</u>—social services, VAW services, criminal justice, health services, religious organizations—and <u>informal</u>—friends, relatives, neighbours [13, 14]. Specifically, facilitators and barriers to access as well as lessons learned will be synthesized. Typically, two main types of disasters exist, natural and technological. The former occurs "outside the control of humans" whereas the latter is due to "breakdown in human-made systems" [45]. Therefore, in this scoping review, we focus on both: (i) the breakdown in human-made systems (i.e., breakdown in financial systems,) that lead to economic recessions and other problems as well as natural environment disasters (i.e., floods, hurricanes, tornados, earthquakes) and infectious

**Table 1. MEDLINE (OVID) preliminary search strategy.**

1. Battered Women/

2. domestic violence/ or spouse abuse/ or gender-based violence/ or intimate partner violence/ or family violence/

3. ((wife or wives or wom#n) adj3 batter*).ti,ab,kf.

4. ((violen* or abus*) adj3 (partner* or wom#n or spous* or wife or wives or marital or marriage*)).ti,ab,kf.

5. ((domestic* or home*) adj3 (violen* or abus*)).ti,ab,kf.

6. ((relation* or interperson*) adj3 (abuse* or violen*)).ti,ab,kf.

7. (violen* adj3 (date* or dating)).ti,ab,kf.

8. (date* adj3 rape*).ti,ab,kf.

9. ((domestic* or marital or partner* or spous*) adj3 rape*).ti,ab,kf.

10. ((domestic* or marital or partner* or spous*) adj3 (sex* adj1 (abuse* or assault*))).ti,ab,kf.

11. (gender-based adj3 (violen* or abus*)).ti,ab,kf.

12. or/1-11

13. stressful life events/ disease outbreaks/ or epidemics/ or pandemics/

14. COVID-19/ or exp COVID-19 Testing/ or COVID-19 Vaccines/ or SARS-CoV-2/

15. coronavirus/ or betacoronavirus/ or coronavirus infections/

16. (disease* adj3 outbreak*).ti,ab,kf.

17. (pandemic* or epidemic*).ti,ab,kf.

18. ((new or novel or "19" or "2019" or Wuhan or Hubei or China or Chinese) adj3 (coronavirus* or corona virus* or betacoronavirus* or CoV or HCoV)).ti,ab,kf,ot.

19. (nCoV* or 2019nCoV or 19nCoV or COVID19* or COVID or SARS-COV-2 or SARSCOV-2 or SARS-COV2 or SARSCOV2 or SARS coronavirus 2 or Severe Acute Respiratory Syndrome Coronavirus 2 or Severe Acute Respiratory Syndrome Corona Virus 2).ti,ab,kf,nm,ot,ox,rx,px.

20. (longCOVID* or postCOVID* or postcoronavirus* or postSARS*).ti,ab,kf,ot.

21. (coronavirus* or corona virus* or betacoronavirus*).ti,ab,kf,ot.

22. ((Wuhan or Hubei) adj5 pneumonia).ti,ab,kf,ot.

23. Economic Recession/

24. Natural disaster/ or environmental disaster/

25. (flood* or earthquake* or tsunami* or tornado* or hurricane* or cyclone* or wildfire* or landslide* or drought* or avalanche* or heat wave* or volcan* or blizzard* or fire* or ice storm*).ti,ab,kf.

26. recession*.ti,ab,kf.

27. (econom* adj3 (depression* or uncertainty or downturn*)).ti,ab,kf.

28. or/13-26

29. health services accessibility/ or health equity/

30. social support/ or community support/ or psychosocial support systems/ or Friends/ or community

31. help-seeking behavior/

32. (health adj3 (service* or care*) adj3 (access* or inaccessib* or unreachable* or unattainab* or reach* or equit* or inequit* or use* or using or seek*)).ti,ab,kf.

33. (healthcare* adj3 (access* or inaccessib* or unreachable* or unattainab* or reach* or equit* or inequit* or use* or using or seek*)).ti,ab,kf.

34. (support* adj3 (access* or inaccessib* or unreachable* or unattainab* or reach* or equit* or inequit* or use* or using or seek*)).ti,ab,kf.*ƒ*

35. ((help* or assistanc*) adj3 (access* or inaccessib* or unreachable* or unattainab* or reach* or equit* or inequit* or use* or using or seek*)).ti,ab,kf.

36. ((famil* or neighbo?r* or relative* or friend*) adj3 (access* or inaccessib* or unreachable* or unattainab* or reach* or equit* or inequit* or use* or using or seek*)).ti,ab,kf.

37. (shelter* adj3 (access* or inaccessib* or unreachable* or unattainab* or reach* or equit* or inequit* or use* or using or seek*)).ti,ab,kf.

38. or/28-36

39. 12 and 27 and 37

This strategy will be appropriately translated to the other databases listed above.

**Table 2. Eligibility criteria.**

| | Inclusion | Exclusion |
|---|---|---|
| Population | Individuals who experienced IPV, aged 13 or older trying to access services during SLEs. | All populations other than individuals 13 or older who experienced IPV and accessed services during SLEs. |
| Phenomenon of Interest/Context | Accessibility to formal and informal services during SLEs in high-income countries. Formal includes healthcare, social, violence against women organizations, spiritual, criminal justice, labour. Informal has family, friends & neighbours. SLEs include pandemics (i.e., COVID-19), disasters/emergencies (i.e., natural environmental), economic recessions. IPV between intimate partners (including domestic violence focusing on intimate relationship between the couple). | Accessibility to services for all other individuals who have NOT experienced IPV during SLEs. Articles about low- or middle-income countries. Articles addressing domestic violence focusing on the child-parent dynamic versus IPV. |
| Types of evidence sources | Qualitative, quantitative, and mixed methods studies<br>• Experimental/quasi experimental<br>• Observational<br>• Policy/Government/Program documents<br>• Relevant reports/thesis. | Literature, Systematic and Scoping Reviews<br>Commentaries<br>Opinion pieces<br>Editorials<br>Conference abstracts |
| Year of publication | No cut-off for date | - |
| Language of publication | English and French | All other languages other than English and French. |

disease crises (i.e., epidemics and pandemics). Please note that we do not use the key terms of 'formal and informal services' as these are academic terms that may not necessarily represent services we are seeking. Rather, they provide a categorical description with a definition of what is meant in the context of this review and the larger study. Acknowledging that males also experience IPV perpetrated by their partners and that the majority of IPV victims are females who experience violence perpetrated by a male partner [46, 47], relevant studies on IPV experienced by women and gender minorities, perpetrated by their partner (regardless of gender) will be retained.

*Context*. For this review, the focus is on high-income countries because the scoping review is part of (and to inform) a larger study looking at access to both informal (family, friends, neighbours) and formal violence against women (VAW) supports for individuals who experienced IPV during COVID-19 lockdowns in Canada. Low and middle-income (LMICs) countries will be excluded as there are other factors that are not relevant to a study conducted in a high-income country like Canada and require different approaches and considerations. The definition of high-income countries as outlined by the World Bank will be used [48]. Although part of a larger study that focuses on COVID-19 lockdown measures, this scoping review is designed to be broader and will include other stressful live including pandemics, natural disasters, and economic recessions in high-income countries.

*Types of evidence sources*. The review will include all research study types investigating the phenomenon (i.e., qualitative, quantitative, and mixed methods) and published in peer-reviewed or grey literature. Evidence sources in the form of published letters, commentaries, opinion pieces, editorials and conference abstracts will be excluded. Systematic, scoping, or other reviews will not be included but the references will be checked for relevant articles. For feasibility, only English and French language studies that explore access to services for victims of IPV during and/or following disasters (i.e., pandemic, economic recession, and natural disasters) in high-income countries will be included. There will be no cut-off for the date as the authors are interested in key historical events (i.e., disasters, emergencies, economic recessions) and the response in terms of service provision for a particular at-risk population (individuals who experience IPV).

## Stage 3: Study screening and selection process

Search results from the databases will be collated in the web-based platform Covidence (Veritas Health Innovation) [49]. Duplicates will be removed before abstract screening. Titles and abstracts will be screened by two reviewers and conflicts resolved through discussion. Articles deemed relevant will go to full-text review and follow a similar method by the same two reviewers. If conflict or disagreement takes place during the abstract of full-text review phases, a third reviewer will be enlisted to resolve the issue. The final list to be included in the scoping review will be confirmed by all reviewers and automated through Covidence to produce a flow diagram for inclusion/exclusion decisions.

## Stage 4: Data charting and extraction

Two reviewers will independently extract the data for all included studies; charting will be guided by JBI Manual for Evidence Synthesis Data Charting tool for Scoping Reviews [50]. Table 3 outlines the key information that will be extracted. This tool will be piloted by the reviewers using three studies, and the extraction parameters will be subsequently adjusted accordingly.

## Stage 5: Collating, analysis/synthesis of evidence and reporting of the results

Simple frequency counts (quantitative data) of the key concepts, populations, and relevant results/outcomes will be conducted. Additionally, a summary of the qualitative data through a descriptive analysis of the content will be provided. The focus is on key findings related to the questions on barriers and facilitators to accessing health and social services by individuals who experience IPV, during stressful life events. Additionally, a summary of the recommendations and lessons learned (i.e., knowledge gained from the process or experience of providing services or trying to access services during SLEs) will be reported. Results will be presented visually through tables or diagrams and narrative descriptions. Table 4 is an example tabular presentation of facilitators and barriers of access to informal or formal support/services by individuals who experience IPV during pandemics, economic recessions, and natural disasters in high-income countries. This table may be revised through ongoing discussion among the extractors as studies are analyzed.

**Table 3. Extraction tool (Peters et al., 2021).**

| | |
|---|---|
| 1 | Authors |
| 2 | Year of publication/study |
| 3 | Origin, country of origin, geography (limited to high income countries as defined by World Bank) |
| 4 | Aim/purpose/objective(s) |
| 5 | Population, sample size, gender |
| 6 | Methodology—study design and approach |
| 7 | Context: Access to services by intimate partner survivors during stressful live events (i.e., pandemics (COVID-19), disasters, emergencies, economic recessions). |
| 8 | Quantitatively or qualitatively outcomes that were measured in the studies to access barriers and facilitators to informal and formal support services. |
| 9 | Key findings related to the scoping review questions (facilitators/barriers to access to social supports; lessons learned; factors/determinants of access during stressful life events; sub-groups at higher risk and their experiences. |

**Table 4. Example tabular presentation of data for the scoping review.**

| | Individuals who experience IPV accessing services | | Service providers delivering service |
| --- | --- | --- | --- |
| | **Formal supports** | **Informal supports** | |
| Facilitators | | | |
| Barriers | | | |
| Lessons learned or Recommendations | | | |

**Quality assessment and risk of bias.** As this is a scoping review, a critical appraisal of the quality of the synthesized evidence is not necessary (i.e., summary measures, risk of bias and certainty of the evidence). This scoping review is not intended to appraise the cumulative body of evidence on access to informal and formal services by individuals who experience IPV. It will, instead, map the evidence on barriers and facilitators to accessing these services during SLEs (i.e., pandemics, disasters, recessions) by at-risk individuals. To minimize errors in assessments of the study selection, analysis, and interpretation, at least two reviewers will be involved in the abstract screening, study selection, data extraction and collation of findings. Additionally, the process of resolving disagreements have been decided in advance and the protocol has been completed and shared to ensure all reviewers have a clear outline of the process. The extraction tool will be piloted using three studies and revised accordingly through review team discussion (i.e., exploration and resolution of disagreement) and documentation of decisions. Additionally, to minimize missing data, the team will include two reviewers who are fully bilingual and therefore both French and English publications will be included in this review. As noted in the study design selection, although relevant reviews (i.e., systematic, scoping, narrative) will not be included, the reference list of included studies will be examined to ensure no key articles were missed. Finally, high-income countries are included because of the relevance of the findings to a larger study being conducted in Canada. There are other health systems and social supports systems considerations for LMICs which are not relevant or congruent to a study being conducted in Canada.

## Discussion

Research has demonstrated the negative impacts of IPV on survivors' health outcomes, the strong association between increased experience of IPV during SLEs, as well as the important role of accessing informal and formal services as a protective factor against the experience of IPV. The recent rise in SLEs, be it the pandemic, increasing number of natural or environmental disasters due to climate change or the recurring financial crises present an opportunity and need to understand better ways to provide service to already isolated individuals who experience IPV.

This review will provide key information on known barriers and facilitators for accessing key services and supports for this marginalized population during SLEs, particularly disasters, health emergencies (i.e., pandemics), and economic recessions. The expected results will be used to (i) create dissemination tools (i.e., fact sheets, presentations) to provide more relevant information to/for service providers to use as information for advocacy or program activities; (ii) inform policymakers regarding accessibility to services during disasters and emergency management; (iii) synthesize recent literature emerging as a result of the COVID-19 pandemic and contribute to existing knowledge, and (iv) inform that the methodology and content of the larger study on accessibility to violence and against women organizations during COVID-19 in Canada.

The intervention is focusing on individuals who experience IPV and anticipate that most of the evidence will focus on women, girls, gender minorities and individuals living in rural/remote regions who are at higher risk of IPV. Studies directed at men will also be included

because the study focus is on access and provision of services during SLEs, and gender is not a limitation. Findings from articles relevant to gender will be discussed should they emerge and show differences.

Authors anticipate that the studies will be heterogeneous and complicated to extract. To mitigate this, the team will work closely during the data extraction phase to ensure that the tool and content analysis are continuously updated as needed. Ongoing communication will take place with the team of screeners and extractors to ensure all are aware of the latest tools and updates.

Any changes in the protocol will be reported in the final manuscript which will be drafted for publication. As this is a scoping review of existing literature (i.e., no human participants were involved in the study), no ethics approval was required. Findings will not only be disseminated through academic journals, but they will also be disseminated to knowledge users who are partners in the larger project (i.e., violence against women organizations and associations).

## Supporting information

**S1 Appendix. PRISMA-P (Preferred Reporting Items for Systematic review and Meta-Analysis Protocols) 2015 checklist: Recommended items to address in a systematic review protocol.**
(DOCX)

## Acknowledgments

The authors would like to acknowledge Karine Fournier, the Head of Reference Services at the University of Ottawa Library for her invaluable guidance while creating and finalizing the search strategy. We would also like to acknowledge Julia Hajjar for proof reading/editing the manuscript.

## Author Contributions

**Conceptualization:** Dina Idriss-Wheeler, Ziad El-Khatib, Sanni Yaya.

**Methodology:** Dina Idriss-Wheeler.

**Project administration:** Dina Idriss-Wheeler.

**Supervision:** Ziad El-Khatib, Sanni Yaya.

**Visualization:** Dina Idriss-Wheeler.

**Writing – original draft:** Dina Idriss-Wheeler.

**Writing – review & editing:** Dina Idriss-Wheeler, Ziad El-Khatib, Sanni Yaya.

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
