## [Decision Letter · Decision Letter 0]

19 Aug 2022

PONE-D-22-16705

Access to support services for individuals who experience intimate partner violence during stressful life events (SLEs) in high-income countries: A protocol for a scoping review

PLOS ONE

Dear Dr. Idriss-Wheeler,

Thank you for submitting your manuscript to PLOS ONE. After careful consideration, we feel that it has merit but does not fully meet PLOS ONE’s publication criteria as it currently stands. Therefore, we invite you to submit a revised version of the manuscript that addresses the points raised during the review process.

We look forward to receiving your revised manuscript.

Kind regards,

Guangyu Tong

Academic Editor

PLOS ONE

Journal Requirements:

Additional Editor Comments:

This protocol has been reviewed by both a scoping review expert and a subject matter expert. Please address the comments accordingly. In particular, the connection between the scope and the instrument/data extraction tool they proposed. The COVID context is emphasized in the writing but almost never appeared in the data collection plan, which needs to be fixed/clarified.

Reviewers' comments:

Reviewer's Responses to Questions

**Comments to the Author**

1. Does the manuscript provide a valid rationale for the proposed study, with clearly identified and justified research questions?

Reviewer #1: Yes

Reviewer #2: Yes

2. Is the protocol technically sound and planned in a manner that will lead to a meaningful outcome and allow testing the stated hypotheses?

Reviewer #1: Yes

Reviewer #2: Yes

3. Is the methodology feasible and described in sufficient detail to allow the work to be replicable?

Reviewer #1: Yes

Reviewer #2: Yes

4. Have the authors described where all data underlying the findings will be made available when the study is complete?

Reviewer #1: No

Reviewer #2: Yes

5. Is the manuscript presented in an intelligible fashion and written in standard English?

Reviewer #1: Yes

Reviewer #2: Yes

6. Review Comments to the Author

You may also provide optional suggestions and comments to authors that they might find helpful in planning their study.

Reviewer #1: Thanks for inviting me to review this manuscript. It is well-written and easy to understand. I only have a few minor comments.

Table 2: The exclusion criteria under “Design” are not types of study designs but the types of research?

From lines 225 to 228, “For this review, the focus is on high-income countries because the scoping review is part of (and to inform) a larger study looking at access to both informal (family, friends, neighbours) and formal violence against women (VAW) supports for individuals who experienced IPV during COVID-19 lockdowns in Canada.” Could the authors clarify a little bit more about the role of “COVID-19” in this review? It has been mentioned multiple times in the background section. From the inclusion and exclusion criteria and data extraction form, I feel that the authors are looking at not only COVID-19 but other types of context, such as Disaster, emergency, and economic recession?

Table 3, item 8, authors mentioned that “a summary of the qualitative data through a descriptive analysis of the content will be provided”. How about quantitative outcomes? Will there be any numerical summaries? It would be great if the authors would have a draft version of data extraction form to share.

Table 3, item 8, “Outcomes that are quantitatively or qualitatively described by the studies as related to access barriers and facilitators to informal and formal support services.” How about “Quantitatively or qualitatively outcomes that were measured in the studies to access barriers and facilitators to informal and formal support services.”

In Table 4, the “Lessons learned” seems to be quite an open-ended summary. I am not a content expert, but I feel it might be very challenging to summarize it concisely as different studies could I have very different “lessons learned”. I can imagine that barriers might be more or less similar across studies, but the lessons learned can be very specific to a particular area, a subpopulation, or under a certain situation, etc. I feel the authors may provide a bit more details about what they are particularly looking for.

Reviewer #2: This is an important area of research and I commend you on this work. Please find some feedback re. each section that may help improve the manuscript.

Abstract: excellent and objectives and methods are clearly outlined.

Introduction: the sentence in line 85 could be better linked to the previous one. It appears to be an add-on. It was great to see the relevant literature being comprehensively and coherently introduced. I recommend that you also explain the theoretical framework that this review will adopt. There is a brief mention of disaster social science. Perhaps this approach to accessing support during SLEs could be adopted and explained further. In relation to the study objectives, something you may want to consider is: how to minority ethnic groups in high income countries access support during SLEs? Of course, this is not compulsory and only a suggestion - due to the various barriers that ethnic minority groups may experience.

Method: In line 180, I think you meant MEDLINE? In Table 1, search term 2, I also recommend you include: family violence. For the 'informal' concepts, also consider including "community". For the context, define 'high-income countries' earlier in the paragraph.

Discussion: Excellent.

7. PLOS authors have the option to publish the peer review history of their article (what does this mean?). If published, this will include your full peer review and any attached files.

Reviewer #1: No

Reviewer #2: **Yes: **Lata Satyen

---

## [Author Response · Author response to Decision Letter 0]

14 Sep 2022

Thank you for the opportunity to submit a revised manuscript. Please see response to reviewer and editor comments below.

Additional Editor Comments:

This protocol has been reviewed by both a scoping review expert and a subject matter expert. Please address the comments accordingly. In particular, the connection between the scope and the instrument/data extraction tool they proposed. The COVID context is emphasized in the writing but almost never appeared in the data collection plan, which needs to be fixed/clarified.

RESPONSE: Thank you for your comment. We have revised the protocol accordingly to ensure the instrument/data extraction tool proposed is properly connected to the scope of the review. We have revised both the context section [p. 12 ; lines 271-280 ], Table 2 eligibility criteria [p. 13 ], and Table 3 [p.14 ].

Reviewers' comments:

4. Have the authors described where all data underlying the findings will be made available when the study is complete?

Reviewer #1: No

RESPONSE: Please note that this is a scoping review protocol and we have indicated that once completed, “Findings will not only be disseminated through academic journals, but they will also be disseminated to knowledge users who are partners in the larger project (i.e., violence against women organizations and associations).” [Page 17 - lines 389-391].

Reviewer #1: Thanks for inviting me to review this manuscript. It is well-written and easy to understand. I only have a few minor comments.

RESPONSE: Thank you for your comment. 

Table 2: The exclusion criteria under “Design” are not types of study designs but the types of research?

RESPONSE: Thank you for your question. We have revised accordingly and changed it to types of evidence sources (as suggested by the JJBI Manual for Evidence Synthesis on page 417). [p13]

Source: Peters MDJ, Marnie C, Tricco AC, Pollock D, Munn Z, Alexander L, et al. Updated methodological guidance for the conduct of scoping reviews. JBI Evid Implement [Internet]. 2021 Mar [cited 2022 Apr 21];19(1):3–10. Available from: http://journals.lww.com/ijebh/Fulltext/2021/03000/Updated_methodological_guidance_for_the_conduct_of.2.aspx

From lines 225 to 228, “For this review, the focus is on high-income countries because the scoping review is part of (and to inform) a larger study looking at access to both informal (family, friends, neighbours) and formal violence against women (VAW) supports for individuals who experienced IPV during COVID-19 lockdowns in Canada.” Could the authors clarify a little bit more about the role of “COVID-19” in this review? It has been mentioned multiple times in the background section. From the inclusion and exclusion criteria and data extraction form, I feel that the authors are looking at not only COVID-19 but other types of context, such as Disaster, emergency, and economic recession?

RESPONSE: Thank you and we agree. We have revised the context section and clarified as suggested. [p.10; lines 271-280]

Table 3, item 8, authors mentioned that “a summary of the qualitative data through a descriptive analysis of the content will be provided”. How about quantitative outcomes? Will there be any numerical summaries? It would be great if the authors would have a draft version of data extraction form to share.

RESPONSE: Please note that Item 8 indicates that both quantitative and qualitative outcomes will be extracted and analyzed. The line “a summary of the qualitative data through a descriptive analysis of the content will be provided” is immediately after we indicate that Frequency counts (quantitative data) of key concepts, population s and relevant results/outcomes that we come across will be conducted. As this is a scoping review and more of an exploratory review method, it is difficult to provide a draft of the data extraction form due to the heterogeneity of what we will come across; once we review the literature, we will summarize the findings accordingly. [p.14/15]

Table 3, item 8, “Outcomes that are quantitatively or qualitatively described by the studies as related to access barriers and facilitators to informal and formal support services.” How about “Quantitatively or qualitatively outcomes that were measured in the studies to access barriers and facilitators to informal and formal support services.”

RESPONSE: Thank you for your suggestion; we have revised the sentence accordingly. [p.14]

In Table 4, the “Lessons learned” seems to be quite an open-ended summary. I am not a content expert, but I feel it might be very challenging to summarize it concisely as different studies could I have very different “lessons learned”. I can imagine that barriers might be more or less similar across studies, but the lessons learned can be very specific to a particular area, a subpopulation, or under a certain situation, etc. I feel the authors may provide a bit more details about what they are particularly looking for.

RESPONSE: Thank you, we agree and have revised the manuscript accordingly. We are looking for lessons learned and recommendations from the experiences of individuals who experienced violence during stressful events attempting to access informal or formal supports, as well as from service providers who delivered services during these times [p. 15; lines 330-332].

Reviewer #2: This is an important area of research and I commend you on this work. Please 

find some feedback re. each section that may help improve the manuscript.

Abstract: excellent and objectives and methods are clearly outlined.

RESPONSE: Thank you for your comment.

Introduction: the sentence in line 85 could be better linked to the previous one. It appears to be an add-on. It was great to see the relevant literature being comprehensively and coherently introduced. 

RESPONSE: Thank you for your comment; we have revised the sentence accordingly. [p.3; lines 97-98]

I recommend that you also explain the theoretical framework that this review will adopt. There is a brief mention of disaster social science. Perhaps this approach to accessing support during SLEs could be adopted and explained further. 

RESPONSE: Thank you for your suggestions. We have added the conceptual frameworks guiding our work. [p. 7; lines 174-186].

In relation to the study objectives, something you may want to consider is: how to minority ethnic groups in high income countries access support during SLEs? Of course, this is not compulsory and only a suggestion - due to the various barriers that ethnic minority groups may experience.

RESPONSE: Thank you for your suggestion and we agree; this has now been incorporated [p.8; lines 205-206]. 

Method: In line 180, I think you meant MEDLINE? 

RESPONSE: Thank you, yes. We have revised from MEDILINE to MEDLINE [p.8; line 220]

In Table 1, search term 2, I also recommend you include: family violence. 

RESPONSE: Thank you. We have revised to include family violence [p. 9, Table 1].

For the 'informal' concepts, also consider including "community". 

RESPONSE: Thank you. We have revised to include community in search [p. 9, Table 1].

For the context, define 'high-income countries' earlier in the paragraph.

RESPONSE: Thank you. We have revised to clarify high-income countries as classified by the World Bank Group; provided a link and reference with the URL for the list [p.6; line 169-170].

Discussion: Excellent.

RESPONSE: Thank you very much for your comment. 

RESPONSE: Thank you. We have revised the manuscript to meet PLOS ONE’s style requirements, including file naming. 

2. Please include captions for your Supporting Information files at the end of your manuscript, and update any in-text citations to match accordingly. Please see our Supporting Information guidelines for more information: http://journals.plos.org/plosone/s/supporting-information

RESPONSE: Thank you. We have revised the supporting information files as suggested (renamed and added a caption at the end of the manuscript after the reference list). [p.22]

3. Please review your reference list to ensure that it is complete and correct. If you have cited papers that have been retracted, please include the rationale for doing so in the manuscript text or remove these references and replace them with relevant current references. Any changes to the reference list should be mentioned in the rebuttal letter that accompanies your revised manuscript. If you need to cite a retracted article, indicate the article’s retracted status in the References list and also include a citation and full reference for the retraction notice.

RESPONSE: Thank you. We have checked for retracted articles and we have none in our reference list.

---

## [Decision Letter · Decision Letter 1]

6 Nov 2022

Access to support services for individuals who experience intimate partner violence during stressful life events (SLEs) in high-income countries: protocol for a scoping review

PONE-D-22-16705R1

Dear Dr. Idriss-Wheeler,

We’re pleased to inform you that your manuscript has been judged scientifically suitable for publication and will be formally accepted for publication once it meets all outstanding technical requirements.

Kind regards,

Guangyu Tong

Academic Editor

PLOS ONE

Additional Editor Comments (optional):

Reviewers' comments:

Reviewer's Responses to Questions

**Comments to the Author**

1. Does the manuscript provide a valid rationale for the proposed study, with clearly identified and justified research questions?

Reviewer #1: Yes

2. Is the protocol technically sound and planned in a manner that will lead to a meaningful outcome and allow testing the stated hypotheses?

Reviewer #1: Yes

3. Is the methodology feasible and described in sufficient detail to allow the work to be replicable?

Reviewer #1: Yes

4. Have the authors described where all data underlying the findings will be made available when the study is complete?

Reviewer #1: Yes

5. Is the manuscript presented in an intelligible fashion and written in standard English?

Reviewer #1: Yes

6. Review Comments to the Author

You may also provide optional suggestions and comments to authors that they might find helpful in planning their study.

Reviewer #1: Thanks for making all the suggested changes. The authors have addressed all my comments. I do not have more comments.

7. PLOS authors have the option to publish the peer review history of their article (what does this mean?). If published, this will include your full peer review and any attached files.

Reviewer #1: No

---

## [Editor Report · Acceptance letter]

11 Nov 2022

PONE-D-22-16705R1 

Access to support services for individuals who experience intimate partner violence during stressful life events (SLEs) in high-income countries: protocol for a scoping review 

Dear Dr. Idriss-Wheeler:

I'm pleased to inform you that your manuscript has been deemed suitable for publication in PLOS ONE. Congratulations! Your manuscript is now with our production department. 

Kind regards, 

on behalf of

Dr. Guangyu Tong 

Academic Editor

PLOS ONE